# Carcass Yields and Meat Composition of Roosters of the Portuguese Autochthonous Poultry Breeds: “Branca”, “Amarela”, “Pedrês Portuguesa”, and “Preta Lusitânica”

**DOI:** 10.3390/foods12214020

**Published:** 2023-11-03

**Authors:** Márcio Meira, Isabel M. Afonso, Rebeca Cruz, Júlio Cesar Lopes, Raquel S. Martins, Jéssica Domingues, Virgínia Ribeiro, Rui Dantas, Susana Casal, Nuno V. Brito

**Affiliations:** 1Escola Superior Agrária, Instituto Politécnico de Viana do Castelo, Rua D. Mendo Afonso, 147 Refóios do Lima, 4990-706 Ponte de Lima, Portugal; marciomeira@ipvc.pt (M.M.); jessicadomingues@ipvc.pt (J.D.); 2CISAS, Escola Superior Agrária, Instituto Politécnico de Viana do Castelo, Rua Escola Industrial e Comercial de Nun’Álvares, 4900-347 Viana do Castelo, Portugal; iafonso@esa.ipvc.pt (I.M.A.); juliocesar@esa.ipvc.pt (J.C.L.); 3Requimte—LAQV, Laboratório de Bromatologia e Hidrologia, Faculdade de Farmácia, Universidade do Porto, Rua de Jorge Viterbo Ferreira, 228, 4050-313 Porto, Portugal; rcruz@ff.up.pt (R.C.); raquel2001martins@gmail.com (R.S.M.); sucasal@ff.up.pt (S.C.); 4AMIBA—Associação dos Criadores de Bovinos de Raça Barrosã, 4730-260 Vila Verde, Portugal; virginia.ribeiro@amiba.pt (V.R.); rui.dantas@amiba.pt (R.D.); 51H-TOXRUN—One Health Toxicology Research Unit, University Institute of Health Sciences, CESPU (Cooperativa Ensino Superior Politécnico e Universitário), CRL, 4585-116 Gandra, Portugal

**Keywords:** poultry, autochthonous breeds, carcass yield, physicochemical properties, nutrient profile

## Abstract

Poultry meat is an important part of the human diet, and the valorisation of autochthonous breeds is a determinant for the sustainability of the rural areas. The increasing demand for niche products demands for better knowledge of the carcass characteristics and meat quality of these local populations. The present study aims to characterise the roosters’ meat from the “Branca” (BR), “Amarela” (AM), “Pedrês Portuguesa” (PP), and “Preta Lusitânica” (PL) breeds. A total of 80 birds (*n* = 20 per breed) between 38 and 42 weeks old were slaughtered. The physicochemical and nutritional composition were determined in the breast and drumstick meat. The meat of the PL breed had a higher (*p* ≤ 0.05) pH value, the AM meat revealed a water-holding capacity (WHC) of superior value and moisture content (*p* ≤ 0.05), while the BR breed had the highest (*p* ≤ 0.05) ash content. On the other hand, it was observed that the PP meat had a higher (*p* ≤ 0.05) yellowness index (b*). The breast meat exhibited, in all breeds, a lower pH value, WHC, redness (a*), and lipid content and greater (*p* ≤ 0.05) lightness (L*), b*, moisture, and ash and protein contents compared to the drumstick. Furthermore, it presented higher (*p* ≤ 0.05) K, P, and Mg contents and a superior proportion of total and sensorial relevant amino acids. Regarding the fatty acid profile, the breed differences were more significant in the drumstick, with the AM breed lipids presenting a higher (*p* ≤ 0.05) percentage of MUFAs in the fat, a lower atherogenic index, and a higher (*p* ≤ 0.05) value for the hypocholesterolemic/hypercholesterolemic fatty acids ratio, while the BR breed lipids presented a higher (*p* ≤ 0.05) percentage of PUFAs and PUFAs/SFAs ratio and a lower n-6/n-3 ratio. Considering the results obtained, it can be concluded that the meat from these breeds is a wholesome dietary option, distinguished by a favourable overall nutritional composition marked by elevated protein content, reduced lipid amounts, and amino and fatty acid profiles with desirable nutritional indices.

## 1. Introduction

Poultry meat production and consumption have grown worldwide during the last decades, imposed mainly by the general perception that this type of meat is healthier than red meat and by its lower price [1,2]. Due to this increased demand, the poultry industry has been compelled to adapt its production strategies, namely, by selecting genetic lines with faster growth rates and higher slaughter weights, in comparison with native breeds, leading to a loss of genetic variability and a consequent lower response to climatic change and disease [3,4,5].

According to the FAO [6], it is estimated that 103 of the 1640 chicken breeds identified worldwide are already extinct, of which 95 were from Europe and the Caucasus. In Portugal, four autochthonous chicken breeds are recognised, “Branca” (BR), “Amarela” (AM), “Pedrês Portuguesa” (PP), and “Preta Lusitânica” (PL), all of which are classified as endangered breeds [2,7].

Over recent years, consumers have exhibited an emerging interest for poultry products deriving from free-range or organic systems [8], and their knowledge and concerns in relation to ethical and environmental practices, animal welfare, and the balanced use of natural resources has increased, leading to a rise in the demand for poultry products obtained through alternative farming methods [1,2,9,10].

Moreover, the valorisation of local traditions and products has boosted consumer’s interest in niche products. Associated with lower productivity and low-input systems, their unique character meets the preferences of today’s consumers, looking for more genuine products with authentic flavours. Most of them recognise that “non-conventional” production methods produce tastier and healthier alternatives, [1,3,11,12]. This positive trend brought more attention and resources to the preservation of local breeds, increasing the number of producers and animals and the interest of the scientific community [13].

Produced in free-range conditions, rooster production has been undertaken for breeding and fattening purposes, slaughtered for gastronomic ends and commonly sold as a whole carcass, the head, neck, feet, and edible viscera (heart, gizzard, liver, and kidneys) included [2,12,14].

The knowledge of the physicochemical properties and nutritional profile of autochthonous poultry meat is an important factor for their valorisation, with an economic added value for the producer. Studies concerning carcass and meat quality traits of the four Portuguese autochthonous poultry breeds are fundamental in order to promote their use and, consequently, the conservation, valorisation, and attractiveness of these breeds. This meat is recognised for its remarkable texture, colour, and flavour and is widely used in the preparation of some characteristic dishes of Portuguese traditional cuisine [2].

The present study intends to evaluate the carcass characteristics, quality, and nutritional composition of the BR, AM, PP, and PL Portuguese autochthonous rooster breeds’ meat. Meat quality and the recognition of its intrinsic value will contribute to the definition of breeding and the marketing strategies of these important genetic local resources.

## 2. Materials and Methods

This study was carried out at the Agrarian Higher School of the Polytechnic Institute of Viana do Castelo (ESA-IPVC). All the procedures described were approved by the organisation responsible for the Animal Welfare of the Polytechnic Institute of Viana do Castelo (ORBEA-IPVC), in accordance with Decree-Law No. 113/2013, of the 7 August reference ORBEA/ESA/IPVC/001/2022.

### 2.1. Sample Size and Animal Management

This study encompassed 80 birds, comprising 20 roosters per breed, obtained through collaboration with AMIBA (Associação dos Criadores de Bovinos de Raça Barrosã) and its affiliated breeders. Each bird was precisely designated as 21 days old by an earmarked identifier, affixed to the anterior right wing’s crease. This identifier bore a unique serial number on one side, linked to the individual bird, while the other side displayed three letters signifying the breed.

All subjects were reared under free-range conditions within the same geographic vicinity, employing a small-scale production system. The housing facilities were of simple, practical, and traditional design, tailored to accommodate the specific number of animals and the nature of the production. The indoor density did not exceed 4 to 6 birds per square meter, and each animal was allocated a minimum outdoor area of 4 square meters.

During the initial 21 days of life, the birds were nourished with a standard commercial starter diet and had unrestricted access to water. Subsequently, they were granted free-range access to pastures and provided ad libitum access to either corn or a blend of various cereals. Additionally, surplus fresh vegetables and farm by-products, such as cabbages, carrots, pumpkins, lettuce, and similar items, were included in their diet whenever available.

### 2.2. Slaughtering and Carcass Traits Evaluations

In adherence to established protocols, the animals were culled at ages ranging from 38 to 42 weeks. For assessing carcass characteristics, the chosen animals were subjected to weighing (referred to as live weight, LW), followed by a fasting period of 12 to 16 h before slaughter. The roosters were mechanically stunned and then killed through manual exsanguination via carotid and jugular incisions, a process taking approximately 30 to 40 s. Subsequently, they were manually plucked after scalding, subjected to a wash, and then underwent weighing (designated as the weight of plucked and bled carcass, CW1). Post evisceration and a 24 h cooling period at 4 °C, two additional weight measurements were taken: one inclusive of the eviscerated carcass weight with all parts (CW2), and another excluding the head, paws, and edible viscera (CW3).

To account for popular dietary customs encompassing the entire carcass (CW2) along with all edible viscera (head, paws, gizzard, heart, liver, and kidneys), an estimate of the weight (edible viscera, EW) was made. The corresponding carcass yields were determined by expressing these weights as percentages of the initial live weight (referred to as CY1, CY2, CY3, and EY, respectively).

### 2.3. Analytical Determinations

The colour was assessed 24 h post-mortem using a CR-400 Minolta colorimeter (Konica Minolta, Osaka, Japan) calibrated with a white standard plate (Y = 84.7, x = 0.3173, and y = 0.3237, No. 11033074, Japan), featuring an 8 mm measuring area, C illuminant, and 2° standard observer. The results were expressed in L*, a*, and b* in the CIELAB colour space [15], averaging five readings per sample on the breast and drumstick muscle’s inner surface.

The physicochemical parameters of 20 breast and 20 drumstick pieces from each breed were analysed in triplicate. Laboratory-prepared meat samples underwent skin and bone removal, 2 min homogenisation (Grindomix GM200, Retsch, Haan, Germany), and were split: one part for pH, moisture, ash, protein, lipid (AOAC 2000 [16]), and fatty acids; the rest freeze-dried for minerals and amino acid quantification.

The pH was measured 24 h post-mortem, for the whole sample, using a digital portable pH meter (HALLO^TM^ FC2022, Hanna Instruments, Eibar, Spain) with a glass electrode calibrated with pH 4.01 and 7.01 buffers (Hanna Instruments, Eibar, Spain) according to ISO 2917:1974 [17].

The water-holding capacity (WHC) of the breast and drumstick muscles was determined by centrifuging approximately 1 g of muscle, placed on filter paper, in a tube at 1500× *g* for 4 min at 4 °C [18]. The water remaining after centrifugation was quantified by oven-drying the samples at 70 °C overnight. The WHC was calculated as the percentage of water retained: (weight after centrifugation − weight after drying)/initial weight × 100.

The weight difference, before and after applying pressure, was used to determine the loss of water (%) in the breast and drumstick muscle [19]; approximately 5 g of muscle was pressed between two filter papers with a pressure load of 2.25 kg for 5 min. The result expressed as a percentage of water lost by the sample was obtained by the difference in weight before and after the method was performed (initial weight − final weight)/initial weight × 100.

The moisture content, following the oven-drying method at 105 °C up to constant weight, was quantified according to the ISO recommended standard 1442:1997 [20]. The ash content was estimated with the incineration (muffle B150, Nabertherm, Lilienthal, Germany) of the samples for 6 h and a temperature of 550 ± 25 °C, according to the ISO 936:1978 method [21]. The contents of the main minerals (phosphorous (P), potassium (K), calcium (Ca), magnesium (Mg), and sodium (Na)) and trace (iron (Fe), zinc (Zn), manganese (Mn), and copper (Cu)) were determined according to the procedure described by Vale et al. [22]. After digestion (DK 42, Heating Digestor, Velp Scientifica, New York, NY, USA), the K, Ca, Mg, Na, Fe, Zn, Mn, and Cu were determined by flame-atomic absorption spectrometry (Perkin Elmer AAnalyst 200, Waltham, MA, USA), while the phosphorous content was determined according to the Ascorbic Acid Standard Method [23] in a UV/VIS spectrophotometer (Thermo Scientific Evolution 60S, Waltham, MA, USA) at 620 nm.

The protein content was analysed using the Kjeldahl method [24] (ISO 937:1978) involving digestion (DK 20, Velp Scientifica), distillation (UDK 139, V. Scientifica), and titration (Titroline 5000, SI Analytics) with a nitrogen factor of 6.25 [24]. The amino acid profile was determined according to the procedure described by Cohen and Antonis [25]. In brief, approximately 10 mg of freeze-dried sample was carefully weighed into a 10 mL glass hydrolysis tube. Subsequently, 3 mL of 6 M HCl, containing 0.5% (*w*/*v*) phenol, was added to the tube, which was then sealed in a nitrogen atmosphere. The acid hydrolysis process was conducted at 110 °C for 24 h, adhering to the established hydrolysis method described in AOAC 982.30 [16]. Following hydrolysis, 200 μL of the resulting hydrolysate was extracted and neutralised using 6 M NaOH, with the final volume adjusted to 1 mL using a 0.1 M borate buffer. Any undissolved particles were effectively removed through ultracentrifugation at 13,000 rpm for 1 min. Amino acid derivatisation was carried out using the AccQ-Tag derivatisation kit (Waters^TM^, Milford, MA, USA). Specifically, 5 μL of the sample hydrolysate was mixed with 5 μL of a 250 μM internal standard (D-norleucine). To this mixture, 70 μL of borate buffer and 20 μL of the derivatization agent were added. The resulting derivatised amino acids were subsequently subjected to an HPLC analysis using a Jasco HPLC system (Japan). This system was equipped with an AccQ-Tag Amino Acids C_18_ Column (4 µm, 150 × 3.9 mm; Waters, Ireland). The analysis employed a fluorescence detector (FP-920, Jasco, Tokyo, Japan) set at λEx = 250 nm and λEm = 395 nm (Gain = 100), as well as a photodiode array (MD-2010 plus, Jasco, Japan) set to spectrum scanning within the range of λ = 210–600 nm, and a column heater (7981, Jones Chromatography, Hesperia, CA, USA) set at 37 °C. Following injection (5 µL), chromatographic separation was achieved using a mixture of three eluents: a patented aqueous buffer (eluent A, Waters^TM^, Milford, MA, USA); milli-Q water (B); and acetonitrile (C). The chromatographic separation proceeded according to the following timeline, 0 min (A, 100%), 0.5 min (A, 99%; B, 1%), 18 min (A, 95%; B, 5%), 19 min (A, 91%; B, 9%), 28 min (A, 83%; B, 17%), 40 min (B, 60%; C, 40%), and 43 min (A, 100%), resulting in a total analysis time of 53 min, with a flow rate of 1 mL/min. Calibration was performed with the Amino Acid Food and Feed Standard Kit (Waters, USA) that contains 21 amino acids, including alanine, α-aminobutyric acid, arginine, aspartic acid, cystine, cysteic acid, glutamic acid, glycine, histidine, isoleucine, leucine, lysine, methionine, methionine sulfone, phenylalanine, proline, serine, taurine, threonine, tyrosine, and valine. Quantification was performed using a five-point calibration curve using D-norleucine as the internal standard.

The determination of the total fat content was carried out using the Soxhlet method (Behr ED from Behr Labor-Technik GmbH, Düsseldorf, Germany), using ether petroleum as the solvent, followed by a gravimetric quantification. The fatty acid profile was assessed in accordance with the methodology outlined by Cruz et al. [26]. The procedure entailed an initial cold extraction of fat using nonhalogenated organic solvents mixtures, followed by acid methylation using 0.4 M methanolic H_2_SO_4_ (60 °C, 3 h), neutralisation with 0.5 M K_2_CO_3_, and a final extraction of the fatty acid methyl esters with heptane, all conducted in adherence to the prescribed guidelines and employing an injection of the external and internal standards. The separation of the fatty acids was accomplished using a Select FAME column (50 m × 0.25 μm i.d., JW, Agilent, Santa Clara, CA, USA), with helium as the carrier gas, and flame-ionisation detection (FID). Quantification was based on the relative percentage of chromatographic areas of fatty acid methyl esters, after calibration of the FID detector with a reference standard solution of fatty acids (TraceCERT, Supelco CRM47885, Merck Life Science, Algés, Portugal). These percentages were then adjusted to the lipid and moisture contents and to obtain the values relative to the weight of the fresh meat. The average proportion of each specific fatty acid was used to compute the summation of saturated fatty acids (SFAs), monounsaturated fatty acids (MUFAs), polyunsaturated fatty acids (PUFAs), and essential fatty acids (EFA), as well as the ratios of PUFAs to SFAs (PUFA/SFA), n-6 to n-3 (n-6/n-3), hypocholesterolemic-to-hypercholesterolemic fatty acids (H/H index), and the atherogenic (AI) and thrombogenic (TI) indices. The H/H ratio was determined based on a formula referenced in Santos Silva et al. [27], while the AI and TI were calculated in line with the methodology described by Ulbricht and Southgate [28].

### 2.4. Data Analysis

Descriptive statistics (mean and standard deviation, SD) were generated for all the variables in the dataset. For the statistical analysis of the results of the carcass and meat quality, an Analysis of Variance (ANOVA) was performed for all the variables considered in this study by using the IBM SPSS Statistics 23.0 for Windows [29], and the comparisons of means were performed by using Tukey’s test. Student’s “t” test was used to determine the significant differences between the breast and drumstick portions. The assumption of normality and homogeneity of variance were previously assessed using the Shapiro–Wilk and Levene’s tests, respectively. All statements of significance were based on testing at the *p* ≤ 0.05 level.

## 3. Results

### 3.1. Carcass Traits Evaluations

The roosters’ carcass characteristics are detailed in Table 1. The results show that the BR breed stands out as the heaviest (*p* ≤ 0.05) and as the one with the best carcass characteristics, followed by the PL breed, in opposition to the AM and PP roosters with the lowest weights. Concerning yields, no significant differences were observed between the breeds.

### 3.2. Meat Quality Parameters

The physicochemical properties and chemical composition of the breast and drumstick roosters’ meat are shown in Table 2 and Table 3, respectively.

The pH, WHC, and yellowness index (b*) values were significantly different between the breeds, showing the AM with the highest (*p* ≤ 0.05) breast pH value, compared to the other breeds, and a superior value for the WHC in relation to the PL breed. The PP breed had the highest (*p* ≤ 0.05) b* values compared to the BR and PL breeds, breast and drumstick, respectively. 

Globally, the breast meat presented, in relation to the drumstick (*p* ≤ 0.05), lower pH and WHC values, as well as redness (a*), and greater lightness (L*) and yellowness (b*) index values.

Concerning the nutritional composition (Table 3), a significant breed effect (*p* ≤ 0.05) was demonstrated on the moisture and ash contents, in both pieces, but not for the protein and lipid contents. The moisture content was higher (*p* ≤ 0.05) in the AM breed in relation to the PP breed, while the ash content was higher (*p* ≤ 0.05) in the BR breed compared to the PP and PL breeds, breast and drumstick, respectively. When compared by pieces, the breast meat exhibited, in all the breeds, higher (*p* ≤ 0.05) moisture, ash, and protein and lower (*p* ≤ 0.05) lipid contents, relative to the drumstick meat.

### 3.3. Mineral Composition

Between the breeds, differences (*p* ≤ 0.05) were found in the Zn (breast) and Mn (drumstick) contents (Table 4). The BR breed exhibited the highest (*p* ≤ 0.05) Zn content compared to the PP breed, while the PP and PL had the highest (*p* ≤ 0.05) Mn content compared to the BR breed. Regarding the pieces, the K, P, and Mg contents were higher (*p* ≤ 0.05) in the breast meat, while the Na, Fe, Zn, Mn, and Cu contents were in the drumstick (*p* ≤ 0.05), regardless of the breed considered.

### 3.4. Amino Acid profile

A total of eighteen amino acids, nine essential amino acids (EAAs) and nine non-essential amino acids (NEAAs), were detected and identified (Table 5). Another amino acid was also detected, but not identified, whose concentration (in histidine equivalents) varied between 1.04–1.54 and 0.95–1.40 mg/100 g of meat, breast and drumstick, respectively. EAAs such as arginine, isoleucine, leucine, lysine, and valine were predominant, while alanine, aspartic acid, glutamic acid, and taurine were the most representative NEAAs. Across both meat pieces, the major amino acid was glutamic acid, followed by aspartic acid, leucine, lysine, arginine, and alanine, representing approximately 55% of the total AAs in the meat.

Unlike the total protein, the AAs profile differed between the breeds, with the PL breed having notably higher (*p* ≤ 0.05) content of both EAAs and NEAAs compared to the other breeds, except for histidine and cysteine in the breast and arginine, histidine, isoleucine, leucine, methionine, phenylalanine, threonine, valine, cysteine, serine, taurine, and tyrosine in the drumstick. Comparing the two meat pieces, the breast meat showed, in all the breeds, higher (*p* ≤ 0.05) protein content, with higher (*p* ≤ 0.05) concentrations of both total EAAs and total NEAAs.

The ratio of EAAs to NEAAs and EAAs to total AAs did not differ significantly between breeds and pieces. The content of AAs that influence taste and flavour perception (alanine, aspartic acid, glutamic acid, glycine, serine, and threonine) was higher (*p* ≤ 0.05) in the breast meat, except in the PP breed, where no differences were observed between the pieces (*p* > 0.05).

### 3.5. Fatty Acid Profile

The fatty acid (FA) profile is shown in Table 6. The oleic (C18:1), linoleic (C18:2n-6, LA), and palmitic (C16:0) acids were the most representative ones. Oleic acid was the main one in all the breeds, particularly in the drumstick, having the highest (*p* ≤ 0.05) proportion in the AM breed fat. In alignment, MUFAs were the most abundant FA, followed by SFAs and PUFAs. There were also breed-specific differences in the drumstick composition, with a higher (*p* ≤ 0.05) percentage of MUFAs in the AM, compared to the BR breed, and PUFAs in the BR breed fat in relation to the AM breed. The meat piece proved to be a source of variation between the main FAs, with a significant (*p* ≤ 0.05) effect on the total proportion of MUFAs (drumstick > breast) and PUFAs (breast > drumstick).

In order to evaluate the four breeds’ meat fat nutritional value, the PUFAs/SFAs, n-6/n-3, and H/H ratios were calculated, as well as the AI and TI (Table 7).

The BR breed showed a higher (*p* ≤ 0.05) PUFAs/SFAs ratio compared to the AM and PP breeds and a lower n-6/n-3 ratio in relation to the AM breed, while the AM breed presented the lowest (*p* ≤ 0.05) value for the AI and the higher (*p* ≤ 0.05) value for the H/H ratio compared to the PP breed. Between the pieces, the PUFAs/SFAs ratio was significantly higher in the breast, in all the breeds, unlike the n-6/n-3 ratio and TI, which were lower (*p* ≤ 0.05).

## 4. Discussion

In recent years, many efforts have been made to understand the factors that affect carcass characteristics and meat quality parameters in the poultry sector. In traditional production systems, recent studies [30,31,32] investigated the productive performance, slaughter yield, and meat quality of different native breeds and showed that these slow-growing chickens can effectively be used in alternative production systems.

A similar weight was observed in the roosters of the four autochthonous Portuguese breeds, compared to other European slow-growing genotypes, slaughtered at similar ages, with values ranging between 2477 and 3230 g [31,32,33,34,35].

The carcass yields obtained were consistent with those reported in previous studies with Portuguese breeds [36]. However, as a consequence of higher muscle mass and a lower proportion of non-edible parts such as the head, feet, feathers, blood, and abdominal fat, a slightly higher CY3 yield was observed compared to other European genotypes [32,34,35]. Moreover, the approximately 10% higher yield of CY2 compared to CY3 means a higher product and profit, which strengthens traditional marketing systems.

Muscle pH has been correlated to most meat quality parameters, such as meat colour, WHC, and tenderness [37]. The differences in pH values between breeds might be explained by the different response to pre-slaughter stress and glycogen reserves at slaughter [38]. All the pH values are within the normal range for 24 h post-mortem measurements [39,40], in agreement with other poultry studies (hens, roosters, and capons) [31,41,42,43], and do not indicate quality defects, such as PSE (pale, soft, and exudative) or DFD (dark, firm, and dry) meat [1]. The pH also depends on the muscle type (oxidative muscles in the drumstick vs. glycolytic muscles in the breast), as the red fibres and their oxidative metabolism have a lower glycogen content, which limits the muscle pH post-mortem drop [31,44].

The WHC can be influenced by age, genotype, pre-slaughter conditions, and the post-mortem evolution of the meat pH [45], which could explain the different values between breeds, as it is known that the susceptibility of animals to stress factors also varies among genotypes [11], as does the ability to store energy in muscles [45]. The lower WHC observed in the breast compared to the drumstick value, which is associated with greater water loss, is related to the muscle pH value [8,40], as low muscle pH and protein denaturation are considered the main determinants of the WHC in meat [46], leading to higher drip and cooking losses [47]. Similar results for WHC were reported by Castellini et al. [40] for organic “Ross” broilers (53.49 vs. 57.45%, breast and drumstick, respectively), slaughtered at 56 or 81 days of age, and Miguel et al. [33] (6.13 vs. 7.81%, breast and drumstick, respectively), for pressing loss in “Castelhana Negra” roosters, slaughtered at 29 weeks of age. Results in the same range were found in males (roosters and capons) of the “Amarela” and “Pedrês Portuguesa” breeds (14–15%) [43] and males (roosters and capons) of the “Mos” breed (7–13%) [42,48] but with a different method (i.e., cooking loss) from the one used in this study (i.e., pressing loss).

The colour of meat is an important factor in its quality, as it is influenced by myoglobin content, pH, and muscle type [8,49]. Consumers use colour and overall appearance to select poultry meat and meat products, with the lightness (L*) and intensity (a* and b*) of the colour being important parameters [32,34]. Our results show that the meat of the four breeds exhibited intense lightness (L*) and values of the redness (a*) and yellowness (b*) index > 0, in agreement with Wattanachant et al. [50] who reported an increase in the L*, a*, and b* values in indigenous chickens compared to broiler chickens. It has been documented that breast and drumstick muscles have different colour characteristics, with breast meat being lighter (L*) and more yellow (b*) but less red (a*) than drumstick meat [42,51], as observed in the present study. Differences in the light-scattering properties of the sarcoplasmic proteins [52] and the uptake of carotenoids and xanthophyll [8,44,53] explain these colour differences. The red fibres of the drumstick muscle, which are rich in myoglobin and haem pigments, explain its higher redness (a*) [31,44]. These results suggest that different autochthonous breeds may share common colour parameters due to similar production and feeding systems [32,42,48]. According to the present results, consumer preference could be focused on the meat of these breeds, overall, with a more intense colour, which could be exploited as a distinctive trait, thus creating a specific market niche.

The moisture content of meat is crucial for its quality, as it influences the perceived juiciness [32]. In this study, the moisture content of the meat from the four different breeds ranged from 73% to 75%, breast and drumstick, respectively, which is consistent with the values reported for other genotypes [30,33,37,42,48]. The authors also found that the drumstick muscle had a higher moisture content than the breast muscle (*p* ≤ 0.05), which may be due to increased muscle activity and tonicity [32].

Ash indicates muscle mineral content. These minerals are associated with the organic compounds involved in the muscle contraction process, and their content increases as the animal grows [54]. Compared to other slow-growing genotypes, with slightly higher or lower values, the differences could be due to the lack of standardisation of productive performance and the meat quality traits of the animals [30,42,48]. Globally, poultry meat is a valuable source of minerals that may vary depending on factors, such as breed, sex, age at slaughter, muscle type, diet and water intake, physiological status, and production system [45,55,56]. Among the minerals studied, K was the most abundant, followed by P and Na, while Fe and Zn were the most significant trace elements, as reported in other genotypes reared in different production systems [57,58]. Our results confirm that muscle type has a clear influence on the mineral content of meat. Similarly, other authors [56,57,58,59,60] found significantly higher contents of K, P, and Mg in the breast comparatively to the drumstick muscle, where the content of trace elements (Fe, Zn, and Mn) is higher. The difference may be explained, at least in part, by differences in blood perfusion and what the blood carries, for example, Fe, Zn, and other trace elements [59].

Adequate K consumption is important to reduce the risk of stroke [61], and the meat of the studied breeds was found to be a good source of this mineral (≈400–450 mg/100 g), contributing to the recommended daily intake (3500 mg/day) [62]. Furthermore, it also provides a significant amount of phosphorus (P) and magnesium (Mg), which are important minerals for several biological processes [55,63,64]. An adequate intake of these minerals is associated with various health benefits, and our results demonstrate that both breast and drumstick meat are good sources of these minerals.

The meat of the four breeds is a valuable source of Fe and Zn, although the amount varies from piece to piece, with it being higher in the drumstick meat. The metabolic capacity and muscle fibre composition are the main factors that affect the amount of Fe in meat, which can explain the different amounts between pieces [58]. Although red meat is the primary dietary source of Fe, consumers often perceive poultry meat as healthier despite its lower Fe content [59,65]. After Fe, Zn is the second most abundant trace element in the human body and plays a crucial role in various biological activities [59,66,67]. As Zn is low in plant-based diets (cereals, legumes, and tubers), including this type of meat in the diet ensures a good part of the recommended daily intake (8 and 11 mg/day for women and men, respectively) [68]. Furthermore, the bioavailability of zinc in meat is higher than in vegetables [67,69]. The differences between pieces can be partly explained by the different composition of muscles and their specific need for Zn to maintain good metabolic functions [59,67]. Although plant foods are the main source of Mn in the human diet [65], considering the average daily intake of meat and its significant importance in the diet, the average values of Mn in the present study contribute to an adequate daily intake (1.8 and 2.3 mg/day for adult women and men, respectively) [68], ensuring its functions in the body.

The breast muscle had significantly higher protein content (24%) compared to the drumstick (20%). The same trend has been observed in other studies with native breeds [31,37,42,48], and this difference has been associated with muscle composition and its ability to deposit protein [70,71]. The protein content of the studied breeds was higher than that reported in commercial lines [57,72,73], including studies under organic and conventional conditions [40,53,74]. Factors such as age, feed management, and production system [41,50,73,75] can affect muscle development and, therefore, the protein deposition in meat [3,8]. These factors may explain the differences in protein content between different studies.

Poultry meat is known for its high nutritional quality and easy digestibility due to its content in essential amino acids [45]. An analysis of the amino acid profile and content is an important factor in determining the nutritional value of meat [76]. Despite its nutritional importance for humans, in native breeds, this parameter has been scarcely described in the literature [3]. Because there are no references to the meat composition of these four Portuguese autochthonous breeds, it was found that arginine, leucine, and lysine are the most abundant essential AAs, while alanine, aspartic acid, and glutamic acid, were predominant in the non-essential fraction, in all the breeds and both pieces, consistent with what was observed in other slow- and fast-growing genotypes, including studies under organic and conventional conditions [3,57,58].

According to the WHO/FAO/UNU, the recommended amount of total protein for an adult human is 0.66 g/kg per day, composed of a daily intake of about 0.18 and 0.48 g/kg of EAAs and NEAAs, respectively. Establishing the specific requirements of EAAs (expressed in mg/100 g) for adults, the consumption of 100 g of meat of any of the breeds ensures a large part of the daily requirement of histidine (1.0), isoleucine (2.0), leucine (3.9), lysine (3.0), methionine (1.0), phenylalanine + tyrosine (2.5), threonine (1.5), and valine (2.6) [77], which can be an important aspect for the valorisation and promotion of the product, given its remarkable quality in EAAs. Furthermore, it is important to mention that all the studied breeds exhibited EAA/TAA and EAA/NEAA ratios higher than the reference values set by the WHO/FAO/UNU of 0.4 and 0.6, respectively [77], which highlights that the meat protein of these breeds is of high quality and can be a valuable option to meet human nutritional needs. On the other hand, the values found for lysine, limiting AAs in cereals and vegetables [78], reinforce the statement that the meat of these breeds is an excellent source of high-quality protein.

AAs are also known to contribute to the taste and flavour attributes of meat, and there is a perception among consumers that meat from native breeds has a better taste and flavour compared to commercial chicken [51]. These differences have been associated with differences in AAs content that contribute to this perception, including alanine, aspartic and glutamic acid, glycine, serine, and threonine, which are known to provide a sweet and umami (savoury) taste to meat [51,79].

All the breeds present considerable contents of alanine, aspartic and glutamic acid, glycine, serine, and threonine. Likewise, other authors [3,50,51,76] have reported higher contents of these amino acids in native breeds when compared to commercial lines, particularly in the glutamic acid content, considered one of the most important amino acids with regard to flavour enhancement. These results may be important indicators for the differentiation of meat compared to commercial lines, ensuring the preference of consumers for meat from autochthonous Portuguese breeds.

Poultry meat is known for its low-fat content, which can be influenced by several factors, including genotype, sex, age, diet, and production system [8,40,74]. The drumstick muscle had higher (*p* ≤ 0.05) lipid content than the breast muscle due to its composition and the type of fibres it contains [41,80]. Red fibres, which are more prevalent in the drumstick, have a higher lipid content compared to white fibres, which are more predominant in the breast muscle as a result of their low need for energy storage [30,31]. Our results are consistent with other studies conducted on males of autochthonous poultry breeds [32,33,48]. However, compared to commercial hybrid breeds and broilers in organic systems, the lipid content in the studied breeds was lower. Commercial breeds such as “Cob”, “Ross”, “Kabir”, “Robusta Maculata”, “817C”, “JA757”, “Bresse”, and “Rhode Island Red” have been reported to have higher lipid content than the studied breeds (0.33–2.37% and 2.47–10.4%, breast and drumstick, respectively) [37,40,57,58,72,73]. The reason why native breeds have low-fat contents, compared to other production systems, is likely due to the increased physical activity that favours myogenesis over lipogenesis [53,58]. This explanation is supported by the results obtained, which may be an important aspect for consumers concerned about diet and fat intake. Likewise, other authors have reported that muscles from slow-growing birds contain less fat than those from commercial lines [3,8].

As previously described, MUFAs were the predominant FA, representing approximately 36 to 42% of the total FA in the meat fat, followed by SFAs (about 31%) and PUFAs (23–27%), different than what was observed in other slow- and fast-growing genotypes, including studies under organic and conventional conditions, in which the SFAs are the predominant fraction with values between 33.96–62.64% and 35.51–66.65%, breast and drumstick, respectively [30,32,40,42,48,50,53,81]. Factors such as age, genotype, sex, muscle type, production system, and diet, mainly, can influence the fatty acid profile [71,81]. It is known that the accumulation of SFAs and MUFAs in poultry muscle depends, partly, on their presence in the feed and their synthesis in the liver [81]. Higher digestion of unsaturated fats reduces the synthesis of SFAs in the liver. However, an increase in PUFAs content can inhibit the action of the ∆9-desaturase enzyme complex, which is the main enzyme responsible for the conversion of SFAs to MUFAs [82]. In line with the general FA composition found in previous studies with Portuguese autochthonous breeds [36,83], the meat of the four breeds contains a good proportion of total and individual PUFAs, particularly EFA C18:2n-6 and C18:3n-3, which are important for human health [71,84]. The present results suggest that local breeds have a good ability to synthesise and/or transfer PUFAs into tissues, making them attractive to health-conscious consumers and economically valuable.

Nutritional indices, including PUFAs/SFAs, n-6/n-3, and atherogenic and thrombogenic indices, are widely used to assess the nutritional value of fat [58,85,86]. The PUFAs/SFAs ratio was found to be higher than the recommended value of 0.45 [87], indicating positive health benefits and a high quality of the meat. However, similar to other studies with native breeds [30,37,42,48], the n-6/n-3 ratio was higher than the recommended value of 4:1, indicating an imbalance of the fatty acid profile. The results suggest that free access to pasture and other vegetable surpluses or by-products from the farms, when available, is not sufficient to reduce the n-6/n-3 ratio, probably due to a predominance of n-6 PUFAs, mainly C18:2n-6, in corn (≈52%) (main diet) and a strong relationship between the dietary fat source and tissue content [76]. However, the observed high values of n-6/n-3 might be compensated by the higher values of PUFAs/SFAs and the favourable H/H ratio (≥2.5) [41,42], used to assess the effect of dietary fatty acid composition on cholesterol metabolism, through the relationship between the hypocholesterolemic FA (C18:1 and PUFAs) and hypercholesterolemic FA (C12:0, C14:0, and C16:0) [87,88]. The present study found that the H/H ratio was higher than that observed in other slow- and fast-growing genotypes [42,58,86,87,89], indicating a higher content of unsaturated fatty acids (UFA) in the meat, which is more beneficial for human health [87,88]. The AI and TI indices are commonly used to assess the nutritional value of foods and their association with the risk of coronary heart disease or cancer. Our results indicate that the AI values of the meat are within the recommended values, below 1.0 (0.33–0.37%, respectively), while the TI values are slightly higher than the recommended value, which should be inferior to 0.5 (0.79–0.89%, respectively) [53,87], similar to other poultry studies [41,42,58,86,87,89]. Overall, the obtained values indicate that the rooster meat from the four Portuguese autochthonous poultry breeds has high nutritional value.

## 5. Conclusions

Although there were no significant differences in terms of slaughter yields, the BR and PL genotypes appear to be the most productive breeds in terms of weight and carcass yield, suggesting they are better suited for meat production, with the other breeds being able to be directed and valued in gastronomy.

Considering the results obtained, it can be concluded that the meat from these breeds is a wholesome dietary option, distinguished by a favourable overall nutritional composition marked by elevated protein content, reduced lipid amounts, and amino and fatty acid profiles with desirable nutritional indices, which are favourable for health-conscious consumers. The significant presence of AAs that impart taste and flavour to the meat further enhances the attractiveness of the product, making it an appetising choice for consumers. These findings are important because they demonstrate their productive potential compared to commercial genotypes and reinforce the idea that their meat can be marketed as a premium product for a specific niche market.

For a comprehensive understanding of the meat’s nutritional profile and to further enhance product differentiation and value, it is essential to conduct studies on meat tenderness and juiciness, as well as detailed sensory analyses. These research efforts will ensure a comprehensive assessment of the meat quality, further strengthening its market position as a tasty, nutritious, and healthy alternative. 

## Figures and Tables

**Table 1 foods-12-04020-t001:** Weights (kg) and carcass yield (%) of the roosters from the four Portuguese autochthonous poultry breeds (mean ± SD, *n* = 20).

	Breeds			
Traits	BR	AM	PP	PL
LW	3.5 ^a^ ± 0.4	2.8 ^b^ ± 0.3	2.8 ^b^ ± 0.4	3.1 ^ab^ ± 0.4
CW1	3.2 ^a^ ± 0.4	2.5 ^b^ ± 0.3	2.5 ^b^ ± 0.4	2.8 ^ab^ ± 0.4
CW2	2.9 ^a^ ± 0.4	2.3 ^b^ ± 0.3	2.2 ^b^ ± 0.4	2.6 ^ab^ ± 0.4
CW3	2.8 ^a^ ± 0.3	2.0 ^b^ ± 0.3	1.8 ^b^ ± 0.4	2.3 ^ab^ ± 0.4
EW	0.33 ^a^ ± 0.04	0.28 ^b^ ± 0.03	0.27 ^b^ ± 0.04	0.29 ^ab^ ± 0.04
CY1	91.5 ^a^ ± 0.8	90.2 ^a^ ± 0.9	90.8 ^a^ ± 1.2	91.3 ^a^ ± 1.8
CY2	83.2 ^a^ ± 2.5	81.5 ^a^ ± 3.0	81.2 ^a^ ± 2.1	82.8 ^a^ ± 2.5
CY3	73.8 ^a^ ± 1.9	71.6 ^a^ ± 3.2	71.2 ^a^ ± 2.0	73.5 ^a^ ± 2.6
EY	9.4 ^a^ ± 0.7	9.9 ^a^ ± 1.2	9.9 ^a^ ± 1.3	9.3 ^a^ ± 0.8

SD—standard deviation; BR—Branca; AM—Amarela; PP—Pedrês Portuguesa; PL—Preta Lusitânica; LW—live weight at slaughter; CW1—bled and plucked carcass weight; CW2—eviscerated carcass, with head, feet, and edible viscera weight; CW3—eviscerated carcass, without head, feet, and edible viscera weight; EW—edible viscera weight (head, feet, gizzard, heart, liver, and kidneys); CY1—bled and plucked carcass yield; CY2—eviscerated carcass yield, with head, feet, and edible viscera; CY3—eviscerated carcass yield, without head, feet, and edible viscera; EY—edible viscera yield (head, feet, gizzard, heart, liver, and kidneys). Lowercase letters within the same row represent significant differences (*p* ≤ 0.05) between breeds.

**Table 2 foods-12-04020-t002:** Physicochemical properties of roosters’ meat from the four Portuguese autochthonous poultry breeds (mean ± SD).

		Breeds			
		BR	AM	PP	PL
Breast (*n* = 20)	pH	5.81 ^bA^ ± 0.09	5.89 ^aA^ ± 0.08	5.83 ^bA^ ± 0.07	5.80 ^bA^ ± 0.10
	WHC (%)	55.9 ^abB^ ± 1.8	56.9 ^aB^ ± 1.9	55.9 ^abB^ ± 2.2	54.2 ^bB^ ± 1.0
	PLoss (%)	12.9 ^aA^ ± 2.2	14.6 ^aA^ ± 1.8	14.8 ^aA^ ± 1.5	14.8 ^aA^ ± 1.1
	Colour				
	L*	50.1 ^aA^ ± 1.9	50.1 ^aA^ ± 2.6	49.8 ^aA^ ± 0.6	49.9 ^aA^ ± 3.0
	a*	4.9 ^aB^ ± 2.0	4.9 ^aB^ ± 1.4	5.3 ^aB^ ± 2.8	4.8 ^aB^ ± 1.5
	b*	11.0 ^bcA^ ± 3.2	11.5 ^abA^ ± 2.1	13.0 ^aA^ ± 3.1	9.5 ^cA^ ± 3.1
Drumstick (*n* = 20)	pH	6.06 ^aA^ ± 0.08	6.04 ^aA^ ± 0.03	5.98 ^bA^ ± 0.05	5.97 ^bA^ ± 0.07
	WHC (%)	56.7 ^abA^ ± 2.0	57.9 ^aA^ ± 1.81	56.3 ^abA^ ± 1.7	55.6 ^bB^ ± 1.7
	PLoss (%)	10.8 ^aB^ ± 2.4	13.2 ^aA^ ± 3.67	12.4 ^aA^ ± 2.1	11.3 ^aB^ ± 1.8
	Colour				
	L*	40.4 ^aB^ ± 2.9	40.0 ^aB^ ± 1.9	40.0 ^aB^ ± 1.7	41.0 ^aB^ ± 2.5
	a*	16.7 ^aA^ ± 1.0	16.5 ^aA^ ± 1.2	16.4 ^aA^ ± 1.6	16.5 ^aA^ ± 0.6
	b*	10.4 ^aA^ ± 1.7	9.9 ^aB^ ± 2.2	11.8 ^bB^ ± 1.9	10.1 ^aA^ ± 2.3

SD—standard deviation; BR—Branca; AM—Amarela; PP—Pedrês Portuguesa; PL—Preta Lusitânica; WHC—water-holding capacity; PLoss—pressing loss; L*—lightness; a*—redness; b*—yellowness. Lowercase letters within the same row represent significant differences (*p* ≤ 0.05) between breeds, while uppercase letters within the same column denote significant differences (*p* ≤ 0.05) between breast and drumstick.

**Table 3 foods-12-04020-t003:** Chemical composition of roosters’ meat from the four Portuguese autochthonous poultry breeds (mean ± SD) (results expressed in % of fresh weight).

		Breeds			
		BR	AM	PP	PL
Breast (*n* = 20)	Moisture (%)	73.40 ^abB^ ± 0.77	73.96 ^aB^ ± 1.14	73.05 ^bB^ ± 0.85	73.64 ^abB^ ± 1.06
	Ash (%)	1.21 ^aA^ ± 0.06	1.15 ^bA^ ± 0.04	1.14 ^bA^ ± 0.06	1.08 ^cA^ ± 0.04
	Protein (%)	24.13 ^aA^ ± 0.68	23.95 ^aA^ ± 0.75	24.49 ^aA^ ± 0.86	24.39 ^aA^ ± 0.92
	Lipids (%)	0.22 ^aA^ ± 0.15	0.20 ^aA^ ± 0.11	0.29 ^aA^ ± 0.24	0.28 ^aA^ ± 0.25
Drumstick (*n* = 20)	Moisture (%)	74.22 ^abA^ ± 1.12	74.75 ^aA^ ± 0.88	73.95 ^bA^ ± 1.07	74.25 ^abA^ ± 0.86
	Ash (%)	1.15 ^aB^ ± 0.06	1.13 ^abA^ ± 0.06	1.10 ^bB^ ± 0.07	1.05 ^cB^ ± 0.04
	Protein (%)	20.12 ^aB^ ± 0.53	20.24 ^aB^ ± 0.53	20.10 ^aB^ ± 0.70	20.39 ^aB^ ± 0.39
	Lipids (%)	1.03 ^aB^ ± 0.47	1.10 ^aB^ ± 0.92	1.15 ^aB^ ± 0.48	0.89 ^aB^ ± 0.29

SD—standard deviation; BR—Branca; AM—Amarela; PP—Pedrês Portuguesa; PL—Preta Lusitânica. Lowercase letters within the same row represent significant differences (*p* ≤ 0.05) between breeds, while uppercase letters within the same column denote significant differences (*p* ≤ 0.05) between breast and drumstick.

**Table 4 foods-12-04020-t004:** Mineral composition of roosters’ meat from the four Portuguese autochthonous poultry breeds (mean ± SD, on a fresh weight basis).

		Breeds			
	Minerals (mg/100 g of Meat)	BR	AM	PP	PL
Breast (*n* = 20)	Macroelements				
	Phosphorous	190.0 ^aA^ ± 13.6	191.8 ^aA^ ± 7.74	194.5 ^aA^ ± 17.4	191.8 ^aA^ ± 13.1
	Potassium	474.9 ^aA^ ± 9.0	482.1 ^aA^ ± 11.94	466.7 ^aA^ ± 26.7	474.6 ^aA^ ± 23.8
	Calcium	12.0 ^aA^ ± 3.0	11.6 ^aA^ ± 3.33	12.2 ^aA^ ± 4.4	10.1 ^aA^ ± 3.7
	Magnesium	33.7 ^aA^ ± 1.4	33.4 ^aA^ ± 1.75	34.7 ^aA^ ± 3.9	34.1 ^aA^ ± 2.9
	Sodium	145.2 ^aB^ ± 7.4	147.0 ^aB^ ± 7.79	145.4 ^aB^ ± 15.2	145.8 ^aB^ ± 10.7
	Trace elements				
	Iron	0.96 ^aB^ ± 0.23	0.98 ^aB^ ± 0.25	1.04 ^aB^ ± 0.24	1.01 ^aB^ ± 0.21
	Zinc	1.49 ^aB^ ± 0.23	1.35 ^abB^ ± 0.18	1.20 ^bB^ ± 0.12	1.33 ^abB^ ± 0.38
	Manganese	0.08 ^aB^ ± 0.02	0.09 ^aB^ ± 0.02	0.09 ^aB^ ± 0.03	0.08 ^aB^ ± 0.03
	Copper	0.10 ^aB^ ± 0.01	0.10 ^aB^ ± 0.02	0.11 ^aB^ ± 0.03	0.10 ^aB^ ± 0.04
Drumstick (*n* = 20)	Macroelements				
	Phosphorous	171.7 ^aB^ ± 10.69	173.4 ^aB^ ± 13.8	171.6 ^aB^ ± 17.9	176.8 ^aB^ ± 13.0
	Potassium	445.0 ^aB^ ± 11.65	450.9 ^aB^ ± 17.8	442.9 ^aB^ ± 28.5	444.9 ^aB^ ± 17.3
	Calcium	12.6 ^aA^ ± 2.94	12.0 ^aA^ ± 3.2	12.7 ^aA^ ± 4.0	10.4 ^aA^ ± 3.5
	Magnesium	28.4 ^aB^ ± 1.04	28.8 ^aB^ ± 1.8	30.3 ^aB^ ± 4.0	29.6 ^aB^ ± 2.1
	Sodium	190.6 ^aA^ ± 9.47	191.7 ^aA^ ± 8.3	190.2 ^aA^ ± 14.6	186.3 ^aA^ ± 9.5
	Trace elements				
	Iron	1.81 ^aA^ ± 0.31	1.74 ^aA^ ± 0.24	1.76 ^aA^ ± 0.21	1.82 ^aA^ ± 0.15
	Zinc	4.27 ^aA^ ± 0.30	4.33 ^aA^ ± 0.66	4.21 ^aA^ ± 0.59	4.16 ^aA^ ± 0.82
	Manganese	0.10 ^cA^ ± 0.02	0.12 ^abA^ ± 0.02	0.13 ^aA^ ± 0.02	0.13 ^aA^ ± 0.02
	Copper	0.14 ^aA^ ± 0.02	0.14 ^aA^ ± 0.03	0.15 ^aA^ ± 0.03	0.15 ^aA^ ± 0.03

SD—standard deviation; BR—Branca; AM—Amarela; PP—Pedrês Portuguesa; PL—Preta Lusitânica. Lowercase letters within the same row represent significant differences (*p* ≤ 0.05) between breeds, while uppercase letters within the same column denote significant differences (*p* ≤ 0.05) between breast and drumstick.

**Table 5 foods-12-04020-t005:** Amino acid profile of roosters’ meat from the four Portuguese autochthonous poultry breeds (mean ± SD, on a fresh weight basis).

		Breeds			
	Amino Acids (g/100 g of Meat)	BR	AM	PP	PL
Breast (*n* = 20)	EAAs				
	Arginine	1.24 ^bA^ ± 0.07	1.16 ^bA^ ± 0.20	1.29 ^bA^ ± 0.27	1.71 ^aA^ ± 0.08
	Histidine	0.56 ^abA^ ± 0.12	0.52 ^bA^ ± 0.08	0.49 ^bA^ ± 0.03	0.70 ^aA^ ± 0.09
	Isoleucine	0.96 ^bA^ ± 0.06	0.83 ^bA^ ± 0.03	1.00 ^bA^ ± 0.19	1.38 ^aA^ ± 0.07
	Leucine	1.60 ^bA^ ± 0.09	1.39 ^bA^ ± 0.05	1.70 ^bA^ ± 0.32	2.22 ^aA^ ± 0.02
	Lysine	1.48 ^bcA^ ± 0.03	1.30 ^cA^ ± 0.05	1.57 ^bA^ ± 0.20	2.18 ^aA^ ± 0.02
	Methionine	0.37 ^bA^ ± 0.05	0.33 ^bA^ ± 0.01	0.37 ^bA^ ± 0.03	0.49 ^aA^ ± 0.03
	Phenylalanine	0.84 ^bA^ ± 0.07	0.77 ^bA^ ± 0.12	0.83 ^bA^ ± 0.16	1.18 ^aA^ ± 0.09
	Threonine	0.46 ^bA^ ± 0.05	0.43 ^bA^ ± 0.06	0.47 ^bA^ ± 0.04	0.66 ^aA^ ± 0.04
	Valine	1.03 ^bA^ ± 0.11	0.93 ^bA^ ± 0.09	1.12 ^bA^ ± 0.22	1.43 ^aA^ ± 0.06
	⅀EAAs	8.54 ^bA^ ± 0.56	7.64 ^bA^ ± 0.55	8.84 ^abA^ ± 1.02	11.95 ^aA^ ± 1.20
	NEAAs				
	Alanine	1.14 ^bA^ ± 0.07	0.99 ^bA^ ± 0.04	1.16 ^bA^ ± 0.23	1.64 ^aA^ ± 0.05
	Aspartic acid	1.54 ^bcA^ ± 0.10	1.32 ^cA^ ± 0.02	1.55 ^bA^ ± 0.15	1.97 ^aA^ ± 0.14
	Cysteine	0.43 ^aA^ ± 0.04	0.34 ^aA^ ± 0.05	0.46 ^aA^ ± 0.08	0.45 ^aA^ ± 0.03
	Glutamic acid	2.46 ^bA^ ± 0.22	2.17 ^bA^ ± 0.07	2.48 ^bA^ ± 0.41	3.28 ^aA^ ± 0.20
	Glycine	0.60 ^bA^ ± 0.05	0.54 ^bA^ ± 0.06	0.67 ^bA^ ± 0.20	0.94 ^aA^ ± 0.07
	Proline	0.57 ^bA^ ± 0.03	0.57 ^bA^ ± 0.06	0.52 ^bA^ ± 0.06	0.83 ^aA^ ± 0.07
	Serine	0.61 ^bA^ ± 0.04	0.52 ^cA^ ± 0.02	0.58 ^bcA^ ± 0.06	0.86 ^aA^ ± 0.03
	Taurine	0.91 ^bA^ ± 0.04	0.83 ^bA^ ± 0.14	0.94 ^bA^ ± 0.22	1.30 ^aA^ ± 0.05
	Tyrosine	0.39 ^bA^ ± 0.04	0.37 ^bA^ ± 0.05	0.45 ^bA^ ± 0.09	0.51 ^aA^ ± 0.03
	⅀NEAAs	8.65 ^bA^ ± 0.47	7.64 ^bA^ ± 0.36	8.80 ^bA^ ± 1.38	11.78 ^aA^ ± 0.46
	n.i.	1.04 ^cA^ ± 0.07	1.22 ^bA^ ± 0.05	1.24 ^bA^ ± 0.10	1.54 ^aA^ ± 0.08
	⅀EAAs/⅀NEAAs	0.99 ^aA^ ± 0.08	1.00 ^aA^ ± 0.04	1.01 ^aA^ ± 0.21	1.01 ^aA^ ± 0.08
	⅀EAAs/⅀TotalAAs	0.50 ^aA^ ± 0.02	0.50 ^aA^ ± 0.01	0.50 ^aA^ ± 0.04	0.50 ^aA^ ± 0.02
	⅀Taste-active amino acids	6.81 ^bA^ ± 0.46	5.97 ^bA^ ± 0.17	6.91 ^bA^ ± 0.80	9.35 ^aA^ ± 0.47
Drumstick (*n* = 20)	EAAs				
	Arginine	1.00 ^abB^ ± 0.11	0.82 ^bB^ ± 0.04	1.21 ^aA^ ± 0.24	1.34 ^aB^ ± 0.20
	Histidine	0.34 ^bcB^ ± 0.07	0.29 ^cB^ ± 0.04	0.42 ^baB^ ± 0.01	0.46 ^aB^ ± 0.06
	Isoleucine	0.87 ^abA^ ± 0.10	0.68 ^bB^ ± 0.03	0.95 ^aA^ ± 0.18	1.02 ^aB^ ± 0.10
	Leucine	1.49 ^abA^ ± 0.18	1.18 ^bB^ ± 0.05	1.65 ^aA^ ± 0.30	1.86 ^aB^ ± 0.22
	Lysine	1.31 ^bB^ ± 0.14	1.06 ^bB^ ± 0.08	1.30 ^bB^ ± 0.18	1.65 ^aB^ ± 0.17
	Methionine	0.32 ^aA^ ± 0.05	0.27 ^aA^ ± 0.05	0.33 ^aA^ ± 0.06	0.34 ^aB^ ± 0.03
	Phenylalanine	0.75 ^abA^ ± 0.11	0.70 ^bA^ ± 0.10	0.90 ^abA^ ± 0.21	1.00 ^aB^ ± 0.15
	Threonine	0.77 ^abB^ ± 0.12	0.60 ^bB^ ± 0.10	0.73 ^abB^ ± 0.09	0.85 ^aB^ ± 0.12
	Valine	0.89 ^abA^ ± 0.14	0.71 ^bB^ ± 0.14	0.98 ^aA^ ± 0.19	1.15 ^aB^ ± 0.13
	⅀EAAs	7.74 ^aB^ ± 0.38	6.30 ^aB^ 1.78	8.47 ^aB^ ± 1.53	9.67 ^aB^ ± 1.77
	NEAAs				
	Alanine	0.99 ^bcA^ ± 0.17	0.89 ^cA^ ± 0.11	1.15 ^bA^ ± 0.15	1.45 ^aB^ ± 0.02
	Aspartic acid	1.28 ^bB^ ± 0.09	1.08 ^cB^ ± 0.08	1.28 ^bB^ ± 0.03	1.66 ^aB^ ± 0.01
	Cysteine	0.33 ^aB^ ± 0.08	0.25 ^aB^ ± 0.04	0.36 ^aB^ ± 0.04	0.35 ^aB^ ± 0.08
	Glutamic acid	2.16 ^bcB^ ± 0.09	1.89 ^cB^ ± 0.18	2.37 ^bB^ ± 0.37	3.02 ^aB^ ± 0.07
	Glycine	0.65 ^bcA^ ± 0.13	0.51 ^cA^ ± 0.06	0.80 ^bA^ ± 0.04	1.03 ^aA^ ± 0.03
	Proline	0.57 ^bA^ ± 0.07	0.54 ^bA^ ± 0.06	0.62 ^bA^ ± 0.11	0.80 ^aA^ ± 0.10
	Serine	0.52 ^bcB^ ± 0.02	0.48 ^cA^ ± 0.06	0.58 ^abA^ ± 0.10	0.80 ^aB^ ± 0.02
	Taurine	0.27 ^abB^ ± 0.05	0.22 ^bB^ ± 0.03	0.29 ^abB^ ± 0.06	0.33 ^aB^ ± 0.07
	Tyrosine	0.30 ^aB^ ± 0.05	0.28 ^aB^ ± 0.05	0.36 ^aB^ ± 0.06	0.40 ^aB^ ± 0.02
	⅀NEAAs	7.07 ^cB^ ± 1.67	6.14 ^cB^ ± 1.08	7.81 ^abB^ ± 2.06	9.84 ^aB^ ± 3.31
	n.i.	0.95 ^cA^ ± 0.07	0.99 ^cA^ ± 0.05	1.24 ^bA^ ± 0.10	1.40 ^aA^ ± 0.08
	⅀EAAs/⅀NEAAs	1.09 ^aA^ ± 0.60	1.03 ^aA^ ± 0.20	1.08 ^aA^ ± 0.46	0.98 ^aA^ ± 1.89
	⅀EAAs/⅀TotalAAs	0.52 ^aA^ ± 0.09	0.51 ^aA^ ± 0.06	0.52 ^aA^ ± 0.07	0.50 ^aA^ ± 0.19
	⅀Taste-active amino acids	6.37 ^bB^ ± 1.48	5.45 ^bB^ ± 1.11	6.91 ^aB^ ± 2.00	8.81 ^aB^ ± 3.16

SD—standard deviation; BR—Branca; AM—Amarela; PP—Pedrês Portuguesa; PL—Preta Lusitânica; EAAs—essential amino acids; NEAAs—non-essential amino acids; n.i.—not identified; ⅀Taste-active amino acids—sum of alanine, aspartic acid, glutamic acid, glycine, serine, and threonine. Lowercase letters within the same row represent significant differences (*p* ≤ 0.05) between breeds, while uppercase letters within the same column denote significant differences (*p* ≤ 0.05) between breast and drumstick.

**Table 6 foods-12-04020-t006:** Fatty acid profile of roosters’ meat fat from the four Portuguese autochthonous poultry breeds (mean ± SD, results expressed as relative percentage).

		Breeds			
	Fatty Acid (%)	BR	AM	PP	PL
Breast (*n* = 20)	C14:0	0.47 ^abB^ ± 0.08	0.40 ^bB^ ± 0.15	0.54 ^aB^ ± 0.10	0.48 ^abB^ ± 0.07
	C16:0	19.83 ^bA^ ± 1.96	19.44 ^abA^ ± 1.72	20.86 ^aA^ ± 1.21	20.68 ^aA^ ± 0.64
	C18:0	9.21 ^abB^ ± 0.69	9.56 ^aA^ ± 0.69	8.91 ^bA^ ± 0.85	9.13 ^abA^ ± 0.69
	Others	1.56 ^abA^ ± 0.35	1.77 ^aA^ ± 0.34	1.46 ^bA^ ± 0.27	1.38 ^bA^ ± 0.26
	⅀SFA	31.1 ^aA^ ± 2.0	31.2 ^aA^ ± 1.8	31.8 ^aA^ ± 1.1	31.7 ^aA^ ± 0.6
	C16:1	2.54 ^abB^ ± 0.63	2.24 ^bB^ ± 0.76	3.01 ^aB^ ± 0.90	2.53 ^abB^ ± 0.55
	C18:1	34.19 ^aA^ ± 2.18	32.82 ^aB^ ± 3.25	32.04 ^aB^ ± 3.15	33.57 ^aB^ ± 3.64
	C20:1	0.31 ^aA^ ± 0.04	0.26 ^abB^ ± 0.08	0.23 ^bA^ ± 0.10	0.27 ^abB^ ± 0.07
	Others	1.49 ^aA^ ± 0.41	1.72 ^aA^ ± 0.51	1.53 ^aA^ ± 0.50	1.43 ^aA^ ± 0.29
	⅀MUFA	38.5 ^aA^ ± 2.1	37.0 ^aB^ ± 3.2	36.8 ^aB^ ± 3.5	37.8 ^aB^ ± 3.9
	C18:2n-6	17.05 ^aB^ ± 3.30	15.71 ^aB^ ± 1.80	16.35 ^aB^ ± 3.49	16.97 ^aB^ ± 2.51
	C20:3n-6	0.28 ^bA^ ± 0.06	0.34 ^abA^ ± 0.13	0.38 ^aA^ ± 0.09	0.31 ^abA^ ± 0.09
	C20:4n-6	6.24 ^aA^ ± 1.74	8.02 ^aA^ ± 2.45	7.32 ^aA^ ± 3.12	6.87 ^aA^ ± 2.47
	⅀n-6-PUFA	23.7 ^aA^ ± 2.2	24.1 ^aA^ ± 2.3	24.2 ^aA^ ± 2.2	24.3 ^aA^ ± 2.1
	C18:3n-3	0.53 ^aB^ ± 0.15	0.36 ^bB^ ± 0.10	0.51 ^aB^ ± 0.21	0.49 ^abB^ ± 0.17
	C22:5n-3	0.81 ^aA^ ± 0.31	0.91 ^aA^ ± 0.37	0.92 ^aA^ ± 0.45	0.83 ^aA^ ± 0.30
	C22:6n-3	0.73 ^aA^ ± 0.21	0.79 ^aA^ ± 0.23	0.84 ^aA^ ± 0.44	0.84 ^aA^ ± 0.33
	⅀n-3-PUFA	2.0 ^aA^ ± 0.5	2.0 ^aA^ ± 0.6	2.2 ^aA^ ± 0.7	2.1 ^aA^ ± 0.6
	⅀LC-PUFAS	8.4 ^aA^ ± 2.2	10.3 ^aA^ ± 3.0	9.8 ^aA^ ± 4.0	9.1 ^aA^ ± 3.0
	⅀PUFA	26.1 ^aA^ ± 2.0	26.5 ^aA^ ± 2.7	26.7 ^aA^ ± 2.5	26.7 ^aA^ ± 2.4
	⅀Trans	0.25 ^aA^ ± 0.07	0.21 ^aA^ ± 0.09	0.23 ^aA^ ± 0.10	0.23 ^aA^ ± 0.09
	EFA	23.8 ^aB^ ± 2.2	24.1 ^aA^ ± 2.3	24.2 ^aA^ ± 2.3	24.3 ^aA^ ± 2.2
Drumstick (*n* = 20)	C14:0	0.58 ^bA^ ± 0.09	0.55 ^bA^ ± 0.10	0.66 ^aA^ ± 0.11	0.61 ^abA^ ± 0.05
	C16:0	19.86 ^bA^ ± 2.08	19.48 ^bA^ ± 1.65	21.34 ^aA^ ± 1.38	20.59 ^abA^ ± 1.28
	C18:0	9.83 ^aA^ ± 0.85	10.14 ^aA^ ± 1.64	9.36 ^aA^ ± 1.26	9.53 ^aA^ ± 0.74
	Others	1.22 ^abB^ ± 0.18	1.35 ^aB^ ± 0.35	1.13 ^bB^ ± 0.08	1.14 ^aB^ ± 0.18
	⅀SFA	31.5 ^aA^ ± 1.8	31.5 ^aA^ ± 2.1	32.5 ^aA^ ± 1.8	31.9 ^aA^ ± 1.6
	C16:1	3.73 ^abA^ ± 0.87	3.50 ^bA^ ± 1.10	4.42 ^aA^ ± 0.94	3.74 ^abA^ ± 0.70
	C18:1	34.71 ^bA^ ± 1.34	37.71 ^aA^ ± 3.88	35.92 ^abA^ ± 3.60	36.90 ^abA^ ± 1.82
	C20:1	0.28 ^bcA^ ± 0.12	0.38 ^aA^ ± 0.05	0.24 ^cA^ ± 0.15	0.35 ^abA^ ± 0.04
	Others	0.95 ^aB^ ± 0.11	0.89 ^aB^ ± 0.22	0.91 ^aB^ ± 0.21	0.83 ^aB^ ± 0.11
	⅀MUFA	39.7 ^bA^ ± 1.8	42.5 ^aA^ ± 4.3	41.5 ^abA^ ± 4.2	41.8 ^abA^ ± 2.2
	C18:2n-6	20.80 ^aA^ ± 2.38	18.59 ^bA^ ± 2.13	18.76 ^bA^ ± 2.16	19.31 ^abA^ ± 1.75
	C20:3n-6	0.20 ^aB^ ± 0.04	0.19 ^aB^ ± 0.08	0.21 ^aB^ ± 0.07	0.20 ^aB^ ± 0.06
	C20:4n-6	3.80 ^aB^ ± 0.75	3.50 ^aB^ ± 1.49	3.32 ^aB^ ± 1.74	3.23 ^aB^ ± 1.23
	⅀n-6-PUFA	24.9 ^aA^ ± 2.2	22.4 ^bA^ ± 3.3	22.4 ^bB^ ± 2.7	22.9 ^abA^ ± 2.4
	C18:3n-3	0.66 ^aA^ ± 0.14	0.46 ^cA^ ± 0.18	0.58 ^abA^ ± 0.10	0.54 ^bcA^ ± 0.10
	C22:5n-3	0.36 ^aB^ ± 0.12	0.28 ^aB^ ± 0.15	0.31 ^aB^ ± 0.22	0.29 ^aB^ ± 0.11
	C22:6n-3	0.36 ^aB^ ± 0.10	0.26 ^bB^ ± 0.07	0.27 ^bB^ ± 0.15	0.32 ^abB^ ± 0.08
	⅀n-3-PUFA	1.35 ^aB^ ± 0.24	0.98 ^bB^ ± 0.32	1.13 ^abB^ ± 0.31	1.13 ^abB^ ± 0.19
	⅀LC-PUFAS	5.05 ^aB^ ± 0.81	4.51 ^aB^ ± 1.59	4.44 ^aB^ ± 2.06	4.31 ^aB^ ± 1.29
	⅀PUFA	26.7 ^aA^ ± 2.4	23.7 ^bB^ ± 3.4	23.9 ^abB^ ± 2.9	24.4 ^abB^ ± 2.5
	⅀Trans	0.28 ^aA^ ± 0.06	0.30 ^aB^ ± 0.13	0.30 ^aB^ ± 0.08	0.31 ^aB^ ± 0.07
	EFA	25.26 ^aA^ ± 2.27	22.55 ^bA^ ± 3.30	22.66 ^bB^ ± 2.72	23.08 ^abA^ ± 2.48

SD—standard deviation; BR—Branca; AM—Amarela; PP—Pedrês Portuguesa; PL—Preta Lusitânica; SFA—saturated fatty acids; MUFA—monounsaturated fatty acids; PUFA—polyunsaturated fatty acids; EFA—essential fatty acids (including linoleic acid, linolenic acid, and arachidonic acid). Lowercase letters within the same row represent significant differences (*p* ≤ 0.05) between breeds, while uppercase letters within the same column denote significant differences (*p* ≤ 0.05) between breast and drumstick.

**Table 7 foods-12-04020-t007:** Nutritional indices of roosters’ meat fat from the four Portuguese autochthonous poultry breeds (mean ± SD, results expressed as % of total fatty acids).

		Breeds			
	Nutritional Indices (%)	BR	AM	PP	PL
Breast (*n* = 20)	PUFA/SFA	0.85 ^aA^ ± 0.11	0.86 ^aA^ ± 0.13	0.84 ^aA^ ± 0.08	0.84 ^aA^ ± 0.07
	n-6/n-3	12.4 ^aB^ ± 3.4	12.5 ^aB^ ± 2.4	11.7 ^aB^ ± 3.3	12.3 ^aB^ ± 3.9
	H/H	3.01 ^abA^ ± 0.46	3.02 ^aA^ ± 0.08	2.76 ^bA^ ± 0.17	2.85 ^abA^ ± 0.10
	AI	0.34 ^abA^ ± 0.05	0.33 ^bA^ ± 0.04	0.36 ^aA^ ± 0.02	0.35 ^abA^ ± 0.01
	TI	0.79 ^aA^ ± 0.07	0.80 ^aB^ ± 0.08	0.81 ^aB^ ± 0.05	0.81 ^aB^ ± 0.02
Drumstick (*n* = 20)	PUFA/SFA	0.85 ^aA^ ± 0.11	0.76 ^bB^ ± 0.13	0.74 ^bB^ ± 0.08	0.77 ^abB^ ± 0.11
	n-6/n-3	19.2 ^bA^ ± 4.0	23.0 ^aA^ ± 4.8	20.8 ^abA^ ± 4.1	20.5 ^abA^ ± 2.2
	H/H	3.04 ^aA^ ± 0.42	3.09 ^aA^ ± 0.37	2.74 ^bA^ ± 0.24	2.90 ^abA^ ± 0.25
	AI	0.34 ^bA^ ± 0.04	0.33 ^bA^ ± 0.04	0.37 ^aA^ ± 0.04	0.35 ^abA^ ± 0.03
	TI	0.83 ^aA^ ± 0.07	0.86 ^aA^ ± 0.08	0.89 ^aA^ ± 0.07	0.86 ^aA^ ± 0.07

SD—standard deviation; BR—Branca; AM—Amarela; PP—Pedrês Portuguesa; PL—Preta Lusitânica; PUFA/SFA—ratio between polyunsaturated and saturated fatty acids; n-6/n-3—ratio between the sum of n-6 and n-3 fatty acids; H/H—ratio between hypocholesterolemic and hypercholesterolemic fatty acids; AI—atherogenic index; TI—thrombogenic index. Lowercase letters within the same row represent significant differences (*p* ≤ 0.05) between breeds, while uppercase letters within the same column denote significant differences (*p* ≤ 0.05) between breast and drumstick.

## Data Availability

The raw data have been submitted to CISAS-IPVC (Center for Research and Development in Agrifood Systems and Sustainability—Instituto Politécnico de Viana do Castelo) and are available on request.

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
