# Peer review of "Carcass Yields and Meat Composition of Roosters of the Portuguese Autochthonous Poultry Breeds: “Branca”, “Amarela”, “Pedrês Portuguesa”, and “Preta Lusitânica”"

_foods, 2023, doi:10.3390/foods12214020_

Round 1

Reviewer 1 Report

Comments and Suggestions for Authors

The aim of the study was to characterize the roosters’ meat from "Branca", "Amarela", "Pedrês Portuguesa"and "Preta Lusitânica" breeds. The subject of the manuscript falls within the general scope of the journal. In my opinion, addressing the following comments:

 Abstract (and throughout the manuscript)

Please enter the level of significance (P<0.05, P>0.05) immediately after: „highest/higher/greater/significant".

 Keywords

Please replace „physicochemical composition” with „physicochemical properties”.

 Introduction

Line 72: Please replace „physicochemical composition” with „physicochemical properties”.

Lines 78-79: I suggest remove this information or use in the Discussion section.

Line 82: „... composition of these four autochthonous roosters´ meat.” Please indicate the breeds names.

 Material and methods

Lines 126-136: I suggest insert this paragraph early (to line 121). Meat was cut and homogenized after pH and color measurements.

Line 139: Please insert a space after number.

Lines 126-136: Please add information about the standard.

Lines 126-136: Why the names of amino acids are written in capital letters?

Line 204: Please add information about the origin of the standard.

 Results

Throughout the section:

Data included in tables shouldn't be cite in the main text.

Please not use word “significantly” when you indicate the p level.

Some results are too generalized. They are not covered by the results contained in the tables. The authors write about higher trait values, but the means are not statistically different.

Throughout the section (and Discussion section): Please check interpretation the WHC results. The higher values (%) not means better WHC (please see the method determination of WHC).

Tables: The letter „a” should be use to mark the highest means. I suggest to mark the means with small letters (x, y) within the same column to indicate significant differences.

Line 240: Please insert „and chemical composition” after „physicochemical properties”.

Line 242: Please remove this sentence.

Lines 258-259: Please indicate higher compared to what.

Lines 269-272: Please remove this paragraph or use in the Discussion section.

Lines 273-274: Please remove this sentence.

Lines 286-287: Please remove this sentence.

Line 288: Please insert „(Table 5)” at the end of the sentence.

Lines 336-337: Please remove this sentence.

Line 336: Please insert „(Table 7)” at the end of the sentence.

Line 338: Please indicate higher/lower compared to what

 Discussion

Line 451-452: I suggest change to: "The breast muscle had significantly higher protein content (24%) compared to drumstick (20%)."

Line 473: Please insert a space between a value and a unit.

Author Response

Please, see  attached reply

Reviewer 2 Report

Comments and Suggestions for Authors

The study is presenting interesting data about the meat quality and the nutritional composition of four indigenous poultry breeds in Portugal. The study is important since the interest in the local breeds that significantly differ from the fast growing broilers is rapidly increasing.

The authors have provided some information in the Introduction as to why the indigenous breeds are now very interesting, however in the context of their study this is rather vague. They have not clearly formulated the aim of the study and the necessity of doing it. Have these four breeds never been studied in regard to meat quality?

The experimental design is generally correct however the authors state that the birds were slaughtered between 38 and 42 weeks of age. How is this age determined, is it specific for each breed? There is few weeks difference and this might influence some of the quality characteristics of the meat.

The methods' description should be improved. It remains unclear if the authors have measured pH on whole muscle of minced meat.

Apart from these remarks, the results are clearly described, they are accompanied with profound discussion and sound conclusions.

Comments on the Quality of English Language

Minor correction of the English language are needed. Please, consult a native speaker for corrections.

Author Response

Please, see attached reply 

Round 2

Reviewer 1 Report

Comments and Suggestions for Authors

The authors have satisfactorily responded to all my questions/comments and made the necessary changes to the manuscript. I recommend an accept the manuscript in its current form.